# A Novel Nanocomposite of Activated Serpentine Mineral Decorated with Magnetic Nanoparticles for Rapid and Effective Adsorption of Hazardous Cationic Dyes: Kinetics and Equilibrium Studies

**DOI:** 10.3390/nano10040684

**Published:** 2020-04-05

**Authors:** Moaaz K. Seliem, Mariusz Barczak, Ioannis Anastopoulos, Dimitrios A. Giannakoudakis

**Affiliations:** 1Faculty of Earth Science, Beni-Suef University, Beni Suef Governorate 621, Egypt; 2Department of Theoretical Chemistry, Institute of Chemical Sciences, Faculty of Chemistry, Maria Curie-Sklodowska University, 20-031 Lublin, Poland; 3Department of Chemistry, University of Cyprus, P.O. Box 20537, Cy-1678 Nicosia, Cyprus; anastopoulos_ioannis@windowslive.com; 4Institute of Physical Chemistry, Polish Academy of Sciences, Kasprzaka 44/52, 01-224 Warsaw, Poland; dagchem@gmail.com

**Keywords:** serpentine, magnetic nanoparticles, adsorption, removal, cationic dyes

## Abstract

A widely distributed mineral, serpentine, obtained from Wadi Ghadir (Eastern Desert in Egypt) was studied as a potential naturally and abundantly available source for the synthesis of an efficient adsorbent for aquatic remediation applications. A novel nanocomposite was synthesized after the exfoliation of the layered structure of serpentine by hydrogen peroxide treatment (serpentine (SP)), followed by decoration with magnetic Fe_3_O_4_ nanoparticles (MNP). The goal behind the utilization of the latter phase was to increase the environmental remediation capability and to incorporate magnetic properties at the final adsorbent, toward a better separation after the use. The fabricated composite (MNP/SP) was characterized by scanning electron microscopy (SEM), Fourier transform infrared spectroscopy (FTIR), and transmission electron microscopy (TEM). The composite’s potential adsorption application toward the removal of two cationic dyes, methylene blue (MB) and malachite green (MG), was investigated. The observed adsorption kinetics was fast, and the highest uptake was observed at pH = 8, with the capacities to reach 162 and 176 mg g^−1^ for MB and MG, respectively, values significantly higher than various other materials tested against these two cationic dyes. Compared to hydrogen peroxide-treated serpentine, the removal efficiency of the composite was higher by 157 and 127% for MB and MG, respectively. The MB and MG were adsorbed because of the favorable electrostatic interactions between MNP/SP active sites and the cationic dyes. The close value capacities suggest that the difference in chemistry of the two dyes does not affect the interactions, with the later occurring via the dyes’ amine functionalities. With increasing ionic strength, the adsorption of the studied basic dyes was slightly decreased, suggesting only partial antagonistic ion effect. The sorbent can be easily regenerated and reused without significant deterioration of its adsorption efficiency, which makes MNP/SP a promising adsorbent for the removal of hazardous pollutants from aquatic environments.

## 1. Introduction

Contamination of the aquatic environments due to the industrial effluents of several products, is considered to be a global challenge. Varied industries including silk, wool, leather, cotton, and paper discharge enormous amounts of hazardous organic water-coloring agents (dyes) in water. The continuous discharge of water–soluble organic dyes, even at low concentrations, into the aquatic system is extremely harmful and dangerous for human beings [1,2,3]. Acute exposure to methylene blue (MB) for instance, can cause increased heart rate, vomiting, shock, cyanosis, jaundice and quadriplegia, and irritation to the skin in humans [4]. Methylene blue (MB), one of the common basic water-soluble dyes, is used for dyeing leather, cotton, printing, calico, tannin, as well as for medicinal purposes [5]. On the other hand, contamination via malachite green (MG), a common cationic dye, was reported to cause carcinogenesis, mutagenesis, and teratogenicity [6]. Several methods with tunable operation conditions were proposed to purify water from organic dyes, including biological treatment, adsorption, coagulation, or precipitation. Among them techniques based on adsorption processes are most frequently used and preferred because of their effectiveness, low cost, and simplicity [7,8]. In general, adsorption of different water contaminates (heavy metals, dyes, anions, and drug residuals) using low-cost materials is a prosperous approach toward a sustainable and green-oriented future. Therefore, different clay-based adsorbents such as kaolin [9], smectite [10], raw and acid-activated bentonite [11], fibrous clay minerals [12], and red mud [13] were used to remove MB from polluted water. On the other hand, bentonite [14], halloysite nanotubes [15], natural zeolite [16], and neem sawdust [17] were used for MG uptake. Moreover, those low-cost natural materials can be further used to prepare composite sorbents with enhanced properties; one of the desired properties of the sorbent is the possibility of its easy separation using contactless magnetic forces. To achieve that the mineral phase can be used to form composite with magnetically active phase, like magnetic Fe_3_O_4_ nanoparticles (MNP). MNP are characterized by high surface area, biocompatibility, great number of active sites, and fast adsorbent separation from solution through magnetic field [18]. After the application of magnetic field to separate the adsorbent, varied desorbing agents can be used to remove the attached pollutants from magnetic nanoparticles and therefore MNP can be reused [19]. The synthetic MNP-coated natural and waste materials were applied significantly in the decontamination of organic pollutants [19,20].

Serpentine (SP, Mg_3_Si_2_O_5_(OH)_4_) is formed by the alteration of Mg–rich silicate minerals such as pyroxene and olivine, producing the three common silicate minerals chrysotile, antigorite, and lizardite. Structurally, SP has an octahedral brucite sheet and two silica tetrahedral sheets. In addition to magnesium and silicon, aluminum, iron, and nickel can be included in the SP composition. Although serpentine is widely distributed in Egypt, eco-friendly, wide-ranging composition, and low-cost, its potential environmental applications related to the removal of hazardous species from waters and wastewaters were not considered. Recently, exfoliation of the 2:1 layered structure of SP by H_2_O_2_ resulted in facilitating the grafting of the cationic surfactant in its structure, which frequently improved the adsorption capacity of SP for Cr(VI) and fluoride [21].

The aim of this work is to synthesize a composite of Fe_3_O_4_ nanoparticles and H_2_O_2_–activated serpentine (MNP/SP) for the removal of cationic dyes (MB and MG) from contaminated aqueous solutions. The effect of different parameters on the uptake capability and the adsorption mechanism were investigated. Finally, kinetics and equilibrium studies were interpreted in the current study.

## 2. Materials and Methods 

### 2.1. Materials 

Herein, serpentine obtained from Wadi Ghadir, Eastern desert in Egypt was subjected to grinding and sieving to get powder with fraction size <100 mesh. The chemicals used were: methylene blue (MB, C_16_H_18_ClN_3_S, Sigma–Aldrich, CAS 122965-43-9, and purity 98.5%), malachite green (MG, C_23_H_25_N_2_, Sigma-Aldrich CAS 123333-61-9, and purity 99%), hydrogen peroxide (30%, H_2_O_2_, Merck, Germany, CAS 7722-84-1), ferric chloride (FeCl_3_·6H_2_O, Loba Chemie, Mumbai, India, CAS 10025-77-1, purity 97%), iron (II) sulfate (FeSO_4_.7H_2_O, Loba Chemie, Mumbai, India, CAS 7782-63-0, purity 99%), ammonia hydroxide solution (NH_4_OH, Loba Chemie, CAS 1336-21-6, purity 25%), and distilled water. To obtain the required pH value, HCl (0.01 M) and NaOH (0.01 M) were used. The molecular structures of studied dyes (MG and MB) are shown in Figure 1.

### 2.2. Preparation of the Composite Adsorbent

Chemical activation of SP sample by H_2_O_2_ was carried out according to the method described previously [21]. Typically, 5 g of SP was added to 50 mL of 50% H_2_O_2_ and stirred for 60 min at 50 °C. In another glass beaker, 1.05 g of FeSO_4_·7H_2_O and 2.1 g of FeCl_3_·6H_2_O were dissolved in 50 mL of distilled water with vigorous stirring for 30 min at room temperature (25 °C). To this mixture, 15 mL of NH_4_OH as a precipitated agent (25%) was added with continues stirring for 30 min. The chemical precipitation technique was applied to obtain magnetic nanoparticles of Fe_3_O_4_ (MNP) under alkaline condition [22]. The contents of the two beakers were mixed together and stirred for 60 min for 50 °C. The resulted magnetic product was separated by magnet before washing by distilled water and ethanol. Then, the solid phase was separated from washing solutions by centrifuge and dried at 70 °C for 24 h. MNP/SP was prepared according to Equation (1) [22]:FeSO_4_·7H_2_O + 2FeCl_3_·6H_2_O + 8NH_4_OH → Fe_3_O_4_ +6NH_4_Cl + (NH_4_)_2_SO_4_ +17H_2_O (1)

The resulting MNP/SP composite was softly grounded and submitted for the characterization and adsorption tests.

### 2.3. Characterizations

Scanning electron microscope (SEM, JSM-6700F, JEOL, Tokyo, Japan) and transmission electron microscope (JEM-2100F, JEOL, Japan) were used to study the morphological features of the examined MNP/SP adsorbent. Fourier transform infrared spectroscopy (FTIR) was used to identify the functional groups (active sites) of the obtained MNP/SP composite using Bruker FT/IR-2000 Spectrometer in the range of 400–4000 cm^−1^. 

The point of zero charge (pH_ZCP_) of the MNP/SP was determined using previously established protocol [23,24]. Total of 0.1 g of MNP/SP composite was added to 50 mL of 0.1 M NaCl solution. The mixture was shaken at 200 rpm for 48 h. The difference between the initial pH and final pH (i.e., pH_f_ – pH_i_) was plotted against pH_i_. The point at which pH_f_–pH_i_ equals zero was the value of zero charge point (pH_ZCP_).

### 2.4. Adsorption Experiments 

First, stock solutions (1000 mg L^−1^) of MB and MG were prepared and the required concentrations were obtained by diluting the prepared stock solutions. In this study, effects of solution pH, contact time, MB and MG initial concentrations and temperature were investigated as factors governing the adsorption process. All adsorption experiments were performed at 25 °C. After separation of liquid phases in all adsorption experiment by centrifuging, a double beam UV-visible absorption spectrometer (Shimadzu, UV 1601) was applied to determine MB (λ_max_ = 660 nm) and MG (λ_max_ = 610 nm) concentrations.

In order to check the influence of pH, 100 mg L^−1^ solutions of MG and MB at different pH values (2.0, 3.0, 5.0, 7.0, 8.0 and 10.0) were prepared. Total of 50 mg of the MNP/SP adsorbent was mixed with 25 mL of the previously prepared MG and MB solutions. The mixtures were shaken at 200 rpm for 2 h using an orbital shaker (SHO-2D). 

MB and MG adsorption kinetics were determined in the adsorption system containing 25 mg of the MNP/SP and 25 mL solution of MB and MG (100 mg L^−1^). The mixtures were agitated at 200 rpm at different contact times (5, 30, 60, 120, 240, and 360 min) and pH = 8.0. The adsorbed amount (*q_t_*) of MB and MG, were calculated from mass balance using the formula:(2)$$q_{t}=\left(C_{0}–C_{t}\right)\frac{\;V}{\;m}$$
where $C_{0}$ (m L^−1^) is the initial concentration of MB (or MG),$\;C_{t}$ is the concentration (mg L^−1^) of the studied organic dyes after time (*t*), V is the volume of the solution (L), and m is the mass of the adsorbent (g). 

Different initial concentrations of MB and MG ranging from 50 to 200 mg L^−1^ were used in evaluation of adsorption equilibrium using 25 mg of the MNP/SP sorbent and 25 mL of the prepared solution. Each adsorption system was shaken at 200 rpm for 120 min at solution pH = 8.0. 

The impact of increasing temperature from 30 to 60 °C on the adsorbed amount of MB and MG was assessed using 25 mg of the MNP/SP and 100 mg L^−1^ of MB and MG concentrations. The adsorption systems at the selected temperatures were shaken at 200 rpm for 2 h. 

All MG and MB adsorption experiments were performed in triplicates and above, and the mean values of the results have been used for data evaluation, with the error to range always below ± 4%.

### 2.5. Fitting of the Adsorption Data

To better describe the adsorption behavior of the dyes onto SP and MNP/SP and to find the possible rate-governing step, the pseudo first-order [25], the pseudo second-order [26], and intra-particle diffusion [27] equations were used in this study (Appendix A). The Langmuir [28] and Freundlich [29] equilibrium adsorption models were also used to fit the adsorption data and details can be found at the Appendix A. 

### 2.6. Salinity Effect on Adsorption

The effect of salinity on MB and MG removal was studied using 50 mL solution of each studied dye with a concentration of 100 mg/L and MNP/SP dose of 50 mg. Different concentrations of NaCl, in the range from 0.2 to 1.2 g L^−1^ were used and all the experiments were performed more than three times.

### 2.7. Reusability – Adsorption/Desorption Cycles

MB and MG desorption experiment was carried out using 100 mL NaOH (0.5 M) as a desorbing agent. The MNP/SP loaded with the studied dyes was continuously agitated on a rotatory shaker at 200 rpm for 2 h at room temperature. The desorption process of MB and MG was repeated five times. At the end of each desorption cycle, the MNP/SP was washed three times by distilled water and dried at 70 °C for 6 h before the next desorption round.

## 3. Results and Discussions

### 3.1. Characterization of the Fe_3_O_4_/activated serpentine (MNP/SP) Adsorbent 

The as-synthesized H_2_O_2_-activated serpentine (SP) and Fe_3_O_4_ with magnetic properties (MNP) are shown in the pictures (Figure 2a,b). The structure analysis by SEM of SP revealed characteristic features as in common serpentine minerals like fibrous chrysotile and platy-like antigorite (Figure 2c). Activation via H_2_O_2_ resulted in partial separation of the layered structure of SP forming opened fractures and pits (Figure 2d). Therefore, in the case of the composite, magnetic iron oxide particles were not only deposited onto the outer SP surface, but also were impregnated in and between the resulted cracks and cavities, forming spherical nanoparticles and aggregations (Figure 2d–f). The existence of these spherical nanoparticles with a high level of homogeneous dispersion, could be expected to enhance the removal of MG and MB because of the increase of the developed MNP/SP composite surface area. The TEM image of SP (Figure 2g) displays the layered structure of a smooth surface. The existence of spherical-like nanoparticles, with diameter of less than 15 nm, anchored on the SP surface can be clearly seen in Figure 2h, testifying further the homogeneous dispersion. 

The chemical composition of the as-synthesized composite was investigated by FTIR spectroscopy and the obtained spectrum is presented in Figure 3. Several absorption band can be attributed to the presence of two phase forming the final composite: 

Since the serpentine is mainly composed of octahedral brucite sheet and two silica tetrahedral sheets, FTIR spectrum displays characteristic bands for both Mg(OH)_2_ and SiO_2_ phase phases. The absorption bands at 1120 and 453 cm^−1^ can be attributed to the asymmetric stretching and bending vibrations of Si-O-Si fragments in silica, respectively. The latter could also be assigned to the stretching vibrations of Mg-O (also Mg-O-Si) fragments of brucite. The band located at 3782 cm^−1^ is due to the presence of isolated silanol groups of silica, while the shoulder band at ~3710 cm^−1^ is due to the stretching vibrations of -OH group in Mg(OH)_2_. Anions present in the interlayer region also give characteristic signals, for example carbonate at 1401 cm^−1^, and nitrate at 990 cm^−1^. The absorption bands at 1621 and 3398 cm^−1^ are due to the bending and stretching vibrations of hydroxy groups of physically absorbed water. Because of high surface heterogeneity, the water is bound to different chemical groups present on the surface, thus apart from the absorption band at 3398 cm^−1^ two other absorption bands corresponding to the stretching vibrations of water are also observed at 3130 and 3032 cm^−1^ [30,31,32]. Fe_3_O_4_ particles and serpentine. The presence of absorption band located at 594 cm^−1^ and its shoulder at ~650 cm^−1^ can be attributed to the bending vibrations associated with the presence of Fe-O (including Fe-O-Si) bonds [33].

### 3.2. Effect of pH on Methylene Blue and Malachite Green Adsorption

The effect of pH on dyes uptake using MNP/SP is shown in Figure 4. Over the selected pH range, the uptake of MB and MG was improved by increasing the pH, showing the maximum removal efficiencies (above 95%) at pH ≥ 8. This behavior can be explained by the presence of different functional groups at MNP/SP interface that can be differently charged with changing pH values. At pH 2-3, the MNP/SP active sites surface become more positive (i.e., protonated functional groups as –OH_2_^+^, Si–OH^+^) because of the high concentration of the competitive H^+^ in solutions [34,35,36]. Thus, the strong electrostatic repulsion between the positively charged MNP/SP active sites and the cationic dyes results in decreasing the MB and MG uptakes. At the solution pH 4–6, the uptake of the dyes is significantly increased. At pH > 7, strong electrostatic attraction between deprotonated groups and the studied dyes improved the adsorption processes greatly. Moreover, the point of zero-charge (pH_ZCP_) was found to be equal to 6.3, indicating that the MNP/SP active sites will bind the dyes more efficiently above this value of pH, because of the favorable electrostatic interactions, which govern the MB and MG adsorption as shown below:(3)MNP/SP−+C16H18ClN3S+→MNP/SP↔C16H18ClN3S (MB uptake)
(4)MNP/SP−+C23H25N2+→MNP/SP↔C23H25N2 (MG uptake)

For these reasons, the adsorption experiments in this work were run in pH = 8. Removal tests were performed also with pure modified serpentine (SP) at different pH and the results are collected in Figure 4 in comparison with the values for MNP/SP. Under the entire pH range, the removal capability for the composite material was more than 2.2-folds higher. At pH = 8, the MB removal was 157% higher for MNP/SP compared to the SP, while for MG 127% higher.

### 3.3. Effect of Contact Time on MB and MG Adsorption

The adsorbed amounts of both dyes as a function of contact time are presented in Figure 5a. Three steps of MB and MG uptake (i.e., steps 1, 2, and 3 representing t < 60 min, 60 < t < 120 min and 120 < t < 360 min, respectively) can be seen, looking at the kinetic data. The first step is very rapid because of the accessibility of great number of MNP/SP active sites for MB and MG adsorption [37]. Step 2 is attributed to the intra-particle diffusion mass transfer reflecting little dyes uptake in this period. In the final step of MB and MG adsorption, the removed amounts of the studied organic dyes were nearly constant in the time range of 120-360 min because of the saturation of MNP/SP active sites by dyes molecules [37].

### 3.4. Adsorption Kinetics and Diffusion Mechanism 

The parameters of the pseudo first-order and the pseudo second-order equations were determined from the linear and non-linear plots (Appendix A) and the obtained results are given in Table 1. Based on the R^2^ values, the pseudo second-order equation (R^2^ > 0.99) fitted well with the MB and MG adsorption processes as compared to the pseudo first-order (*R^2^* < 0.95). Moreover, the calculated *q_e_* obtained from the pseudo-second order fitting is very close with the experimental *q_e_* values confirming the validity of this model in fitting the MG and MB data. The fitting with the intra-particle diffusion model is shown in Appendix A. During all periods of interaction times, the plot was divided into three lines. The first sharp line could be linked to the external mass transfer of MG and MB from the contaminated solution to the external active sites of the MNP/SP composite [38]. The lines 2 and 3 reflected the control of pore-diffusion and saturation stages, respectively [30]. The non-linear plots of the pseudo first-order and the pseudo second-order kinetic models of MB and MG are presented in Appendix A and the obtained parameters are listed in Appendix A.

### 3.5. Adsorption Isotherms

Based on Langmuir assumption, MB and MG adsorption occurs at similar active sites of the prepared MNB/SP composite. The $q_{\max}\;$and b parameters of Langmuir model were calculated from the relation between of *C_e_/q*_e_ and *C*_e_ (Appendix A) and the results are given in Appendix A. The dimensionless separation factor ($R_{s}$) of the Langmuir isotherm model was determined from the following equation [38].
(5)$$R_{s}=\frac{1}{1+k_{L}\;C_{0}}$$

The calculated $R_{s}$ values are in the range 0.05–0.09 for MB and MG indicating favorable adsorption processes of both dyes.

According to Freundlich equation, uptake of MB and MG takes place at heterogeneous active sites forming more than one layer (multilayer) onto the MNP/SP surface. Indeed, apart from monolayer formation governed by electrostatic interactions, some part of the dyes can be adsorbed by π–π interactions between the dyes molecules [36]. The parameters of Freundlich model (K_F_ and n) were determined (cf. Appendix A) from the linear relation between log $q_{e}$ versus log $C_{e}$ (Appendix A). The relation between $C_{e}$ and $q_{e}$ was used in fitting the MB and MG experimental data to the non-linear forms of Langmuir and Freundlich models and the obtained parameters are given in Appendix A. Based on R^2^ values determined for the two models (cf. Appendix A), it can be concluded that adsorption of MB and MG dyes can be fitted using both of them. 

The most important outcome that can be derived from all the above presented and discussed results is that the removal efficiency of the composite is almost equal for both dyes. Considering the molecular weight of MB (319.9 mol g^−1^) and MG (364.9 mol g^−1^), we can suggest that specific sites are responsible for the adsorption. Comparing the molecular structures (cf. Figure 1), it can be also suggested that the size and chemical differences does not affect the interactions involved in adsorption, since adsorption is driven by the interactions of the terminal amine groups located on the edges of the dyes molecules, thus equally accessible for both dyes regardless of their molecular sizes.

### 3.6. Comparison of the MNP/SP Adsorbent with Other Studies

Table 1 collects the values of maximum adsorption capacities of MB and MG by the as-synthesized MNP/SP of this study as compared to natural and modified materials. Based on the $q_{\max}$ values of MB (201 mg g^−1^) and MG (218 mg g^−1^), the MNP/SP sorbent is measured as a promising composite for the removal of the selected cationic organic dyes (MB and MG) from the contaminated solutions. 

### 3.7. Temperature Effect on MB and MG Adsorption 

At 25 °C, the removed amounts of MB and MG are 84 and 88 mg g^−1^, respectively. With increasing the solution temperature to 60 °C, the adsorbed amounts decreased to 70 and 73 mg g^−1^ for MB and MG, respectively. The observed decrease testifies to the exothermic nature of adsorption processes occurring during uptakes of the dyes [45].

### 3.8. Effect of Salinity on MB and MG Adsorption

Dyeing method is associated with increasing salt concentration and therefore, the ionic strength effect on the MG and MB adsorption onto the MNP/SP was investigated. It was observed that with an increase in the salt concentration from 0.2 to 1.2 g L^−1^, the removed amounts were decreased from 88 to 73 mg g^−1^ in case of MG and from 84 to 71 mg g^−1^ in case of MB (Figure 5c). This may be due to the screening effect of salt ions, which resulted in suppressing the electrostatic attractions between MNP/SP adsorbent and the dyes molecules [46]. Furthermore, this decrease in the uptake values can be related to the occupation of some MNP/SP active sites by sodium ions.

### 3.9. The Reusability Studies

Possibility of the multiple use of any sorbent is one of the most important issues determining the possibility of its real use in adsorption-based industrial processes [47,48,49,50,51,52]. Herein, five adsorption/desorption cycles were run to evaluate the possibility of regeneration and reusing of the synthesized composite sorbent. The results are presented in Figure 5d. As it can be seen the regenerated sorbent maintained high sorption retention effectiveness after five cycles toward both dyes. The adsorption capacity of the sorbent after four adsorption/desorption cycles is 73% and 78% for MB and MG, respectively. The decrease of sorption capacities for all the samples can be linked with two possible factors: (i) Irreversible adsorption of some amount of the dyes that could not be desorbed using 0.5 NaOH and (ii) gradual elimination of porosity and surface chemistry upon processing (including water-driven processes occurring during sorption cycles and NaOH-driven during desorption cycles). Nevertheless, the observed retention of adsorption effectiveness toward MB and MG after five cycles clearly showed the applicability of the obtained sorbent.

## 4. Conclusions

A new composite of activated serpentine and magnetic iron oxide nanoparticles was successfully fabricated, characterized, and tested as adsorbent of two cationic dyes (methylene blue and malachite green). The sorbent exhibited the maximum adsorption uptake at pH = 8. Kinetic data revealed that the equilibrium was reached after 120 min of shaking time and the adsorption rate followed the pseudo second-order kinetic model. The obtained sorbent could adsorb as much as 162 mg g^−1^ of MB and 176 mg g^−1^ of MG. The sorbent could be easily regenerated using diluted NaOH solution and reused many times while maintaining over 70% effectiveness after five adsorption cycles. This study clearly shows that serpentine-derived composite materials can be successfully used for fast, efficient, and cost-effective adsorptive removal of hazardous dyes from waters and wastewaters. Moreover, the interactions with the adsorbent surface occur via the cationic amine functionalities, since the adsorption-desorption capability was found similar for both MB and MG.

## Figures and Tables

**Figure 1 nanomaterials-10-00684-f001:**
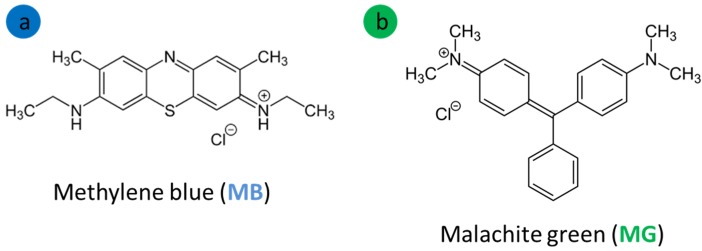
The molecular structures of the studied basic dyes.

**Figure 2 nanomaterials-10-00684-f002:**
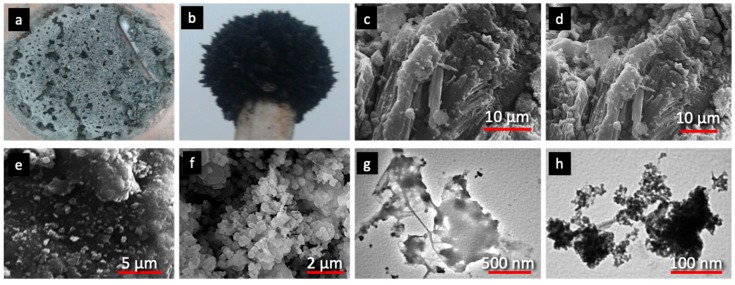
Photographs of serpentine (SP) (**a**) and magnetic Fe_3_O_4_ nanoparticles (MNP) (**b**), scanning electron microscopy (SEM) images of SP (**c**) and MNP/SP (**d**–**f**), transmission electron microscopy (TEM) images of SP (**g**) and MNP/SP (**h**).

**Figure 3 nanomaterials-10-00684-f003:**
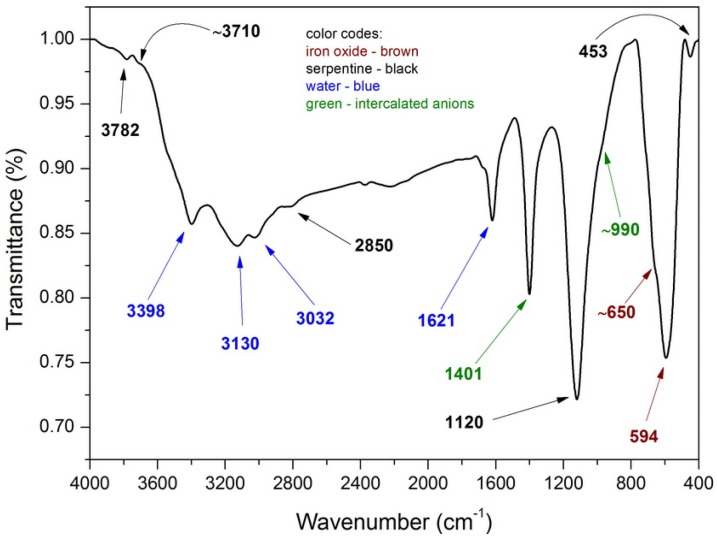
Fourier transform infrared spectroscopy (FTIR) spectrum of MNP/SP composite.

**Figure 4 nanomaterials-10-00684-f004:**
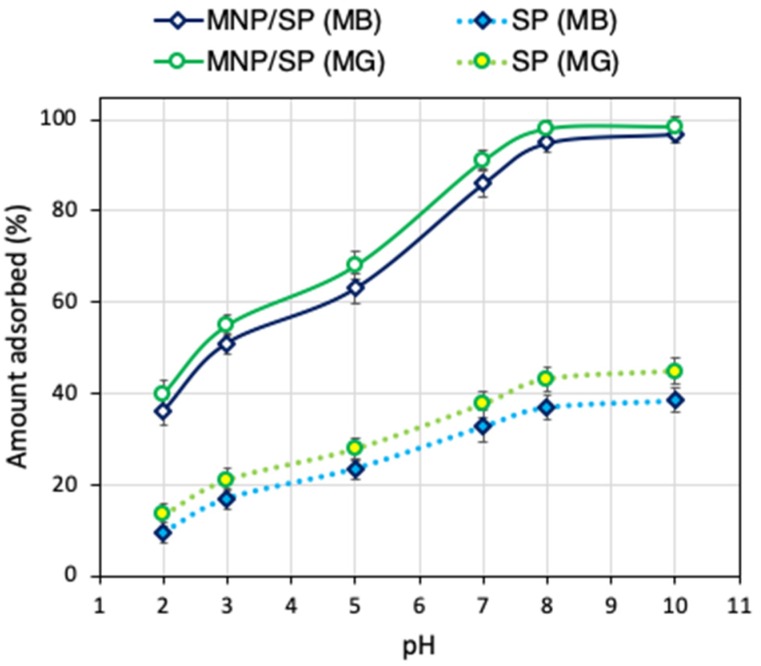
Effect of pH on adsorption of methylene blue (MB) and malachite green (MG) by H_2_O_2_-activated serpentine (SP) and iron oxide modified SP (MNP/SP).

**Figure 5 nanomaterials-10-00684-f005:**
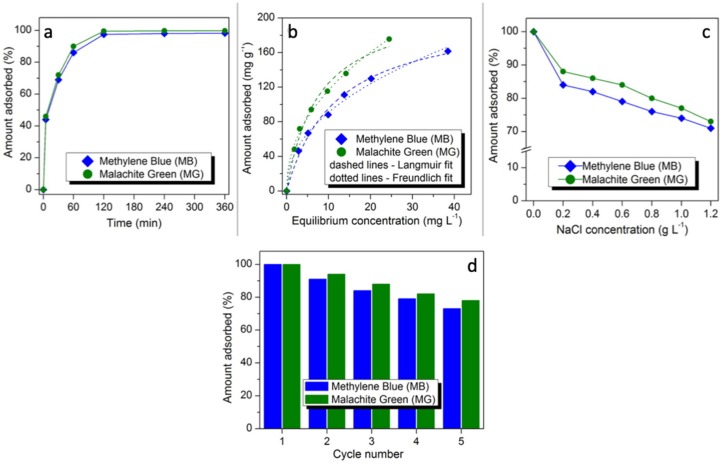
Effect of contact time on adsorption of dyes (**a**), adsorption isotherms with their fitting to Langmuir and Freundlich models (**b**), effect of NaCl addition on adsorption of dyes (**c**), relative adsorption uptakes after sorbent regeneration (**d**). Please note that fitting data are given in the Appendix A.

**Table 1 nanomaterials-10-00684-t001:** Comparison of adsorption capacities (based on Langmuir model) for different materials reported in the literature and the obtained MNP/SP material.

Adsorbate	Adsorbent	Sorption Capacity (mg g^−1^)	Reference
MB	Montmorillonite	64	[20]
MB	Fe_3_O_4_/montmorillonite	106	[20]
MB	Fibrous clay minerals	39–85	[12]
MB	Mn-doped mesoporous MCM-41 silica	132	[39]
MB	Purified diatomite	105	[34]
MB	Chitosan/magnetic silica	201	[40]
MB	Fe_3_O_4_/serpentine composite	201	Current study
MG	Bentonite	179	[14]
MG	Activated carbon	57	[41]
MG	Natural zeolite	24	[16]
MG	Biocarbon prepared from plant root	8	[42]
MG	Halloysite nanotubes	100	[15]
MG	Modified rice husk	12	[43]
MG	Degreased coffee bean	55	[44]
MG	Fe_3_O_4_/serpentine composite	218	Current study

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
