# Peer review of "A Novel Nanocomposite of Activated Serpentine Mineral Decorated with Magnetic Nanoparticles for Rapid and Effective Adsorption of Hazardous Cationic Dyes: Kinetics and Equilibrium Studies"

_nanomaterials, 2020, doi:10.3390/nano10040684_

Round 1
Reviewer 1 Report
Article Name: A new composite of Fe3O4 nanoparticles/H2O2‒activated serpentine for rapid and effective adsorption of hazardous cationic dyes:
Kinetics and equilibrium studies
Recommendation: Accept with minor revision
Reviewer comments:
1) The authors submitted the draft version of the manuscript without proper proofreading. Font mismatch, submitting same table twice (Table 1), strikethrough lines and asking informal questions (lines 277-280) are some of the examples of lack of professionalism.
2) The authors did not explain why the uptake capacity of the adsorbents were different for MB and MG?
3) Did the authors do any regeneration study for the adsorbent after the adsorption?
4) For adsorption isotherm, can the author provide any reference that an adsorption process can follow both Langmuir and Freundlich isotherm model? This is very unusual.
5) What does the fitting of the adsorption process to the pseudo-second order kinetic model imply? What is the significance of that study or findings?
Reviewer 2 Report
I guess a weird version of the original word document has been uploaded. The font size is changing throughout the manuscript and probably the track changes have remained turned on. In line 277, there is a whole sentence deleted before and it remained there. Honestly it is really hard to review a manuscript with such poor quality of styling. Authors must be more careful when uploading manuscripts like this. Some of the figures and equations are not reaching the level of a scientific publication. In my opinion some part of the abstract should be moved to the methods section and the overall message should be clarified why this study was carried out.
Reviewer 3 Report
The paper describes the synthesis new composite based on serpentine and incorporated Fe3O4 nanoparticles. Magnetic composites arouse enormous interest for some environmental applications as they combine useful magnetic properties, low toxicity and the sorption or adsorption properties. The modulation of their properties through the use of different molecules and micro-or nanoparticle has been the subject of numerous research works during the last years. Therefore, the article focuses on research topics of significant relevance at present. There are so many techniques used in this work but have not been discussed adequately. There are some major critical concerns which need to be addressed.
1) I think FTIR studies are not enough to study composition of new material. Since the focus of this study is the composite of Fe3O4 nanoparticles and serpentine, the EDX or XPS studies seem more relevant for such study.
2) SEM and TEM results should be more clearer. Please complete the results with a histogram showing the size distribution of nanoparticles (or sub micro-particles).
3) There is no statistical description in the section on adsorption studies
Apart from the above important concerns, there are some minor issues also. The manuscript needs to be thoroughly checked for incomprehensible statements at many places.
Reviewer 4 Report
The authors have prepared a new sorbent material, by activating serpentine with H2O2 and then adding magnetic iron oxides prepared by co-precipitation. Although this is a relevant topic, the manuscript needs extensive revision before being considered for publication. A list of comments follows:
-Introduction: The authors should justify better the choice of serpentine as substrate. The state of the art about the use of serpentine as adsorbent should be improved and completed. Furthermore, what was the purpose of modifying serpentine with iron oxide nanoparticles? Typically, the aim is to develop magnetic sorbents that could be easily separated from treated water. Nevertheless, nothing is mentioned in the introduction about magnetic separation.
-Scheme 1- authors must improve the resolution of the images (formulas of MB and MG)
-preparation of magnetic SP: The material MNP/SP was separated using centrifugation, why? If the material is magnetic it can be separated using a magnet. Using centrifugation how can the authors ensure that all the sorbent materials are magnetic? If some serpentine is not attached to the magnetic phase it will be also recovered by centrifugation.
-Adsorption experiments: lines 110-11. What is the point of preparing a magnetic sorbent if the, when performing the adsorption experiments, the sorbent is separated by centrifugation?
-Characterization of MNP/SP: The characterization data presented by the authors are not sufficient. The authors must present experimental evidence of the formation of Fe3O4, for example by XRD diffraction. FTIR is not enough to justify the formation of magnetite. What is the average size of the nanoparticles of iron oxide? The scale in the electron microscopy images is not visible. In the FTIR the authors should include in the same plot the FTIR spectrum of the serpentine before adding the magnetic phase. Besides, authors should include the results of the magnetic properties of the MNP/SP (magnetization curve).
- Analysis of the results: The fitting to the kinetics and isotherms models must be performed using the non-linear equations, through non-linear regression. It is not recommended to use the linear form of these equations to estimate the kinetic and equilibrium sorption parameters. For example, the following manuscript (Doi: 10.1021/acs.iecr.7b04724) alerts for this problem in the case of the pseudo-second order kinetic equation. For the analysis of the goodness of the fit, the authors must include the residual plot as well. The comparison of the R2 is not enough.
-Figure 2d is distorted.
- In the results of the adsorption, a comparison with the adsorptive behavior of the serpentine before adding the magnetite (but treated with H2O2) is necessary. How can the authors understand if the adsorption behavior of MNP/SP is due to the active sites of the serpentine or due to the active sites of the magnetic iron oxides?
- Can the material MNP/SP be regenerated and reused in the adsorption experiments?
Round 2
Reviewer 2 Report
I can understand now why this study was carried out and what could be the potential in there. I can also see how much better would it be if the authors would really re-structure the manuscript and emphasize it's strength. A good example is the first sentence of the abstract where the authors describe how the Fe3O4 nanoparticles were made instead of why? Why was it made and why is it better than current solutions? Nowadays supplementary material is acceptable in most of the journals so there is no need to include 12 equations in the manuscript so please simplify. Nanomaterials have a microsoft word template available and I would suggest to use it so the font size wouldnt be different in the different sections. Currently the science in this manuscript is reaching the level of Nanomaterials although its styling, english and clarity is far away.
Author Response
Reply to reviewer
